# An Integrative Approach to Assessing Diet–Cancer Relationships

**DOI:** 10.3390/metabo10040123

**Published:** 2020-03-25

**Authors:** Rachel Murphy

**Affiliations:** 1Cancer Control Research, BC Cancer, 2-107 675 West 10th Avenue, Vancouver, BC V5Z 1L3, Canada; Rachel.murphy@ubc.ca; Tel.: +1-604-822-1397; 2School of Population and Public Health, University of British Columbia, 167-2206 East Mall, Vancouver, BC V6T 1Z3, Canada

**Keywords:** metabolomics, multi-omic, epidemiology, systems biology, integration

## Abstract

The relationship between diet and cancer is often viewed with skepticism by the public and health professionals, despite a considerable body of evidence and general consistency in recommendations over the past decades. A systems biology approach which integrates ‘omics’ data including metabolomics, genetics, metagenomics, transcriptomics and proteomics holds promise for developing a better understanding of how diet affects cancer and for improving the assessment of diet through biomarker discovery thereby renewing confidence in diet–cancer links. This review discusses the application of multi-omics approaches to studies of diet and cancer. Considerations and challenges that need to be addressed to facilitate the investigation of diet–cancer relationships with multi-omic approaches are also discussed.

## 1. Introduction

### Diet and Cancer

Diet is one of the few ubiquitous and potentially modifiable risk factors for cancer. The link between diet and cancer was first noted by Doll and Peto [1] who proposed that approximately 35% of cancers (ranging between 10%–70% depending on the type of cancer) in the United States may be attributable to diet. More contemporary estimates suggest that over 5% of new invasive cancer cases in the United States can be attributed to diet-related factors [2]. Following the work of Doll and Peto [1], the World Cancer Research Fund/American Institute of Cancer Research (WCRF/AICR) published the first global report on diet and cancer in 1997 [3]. The second WCRF/AICR report subsequently provided the world’s largest and most comprehensive assessment of the evidence base on the role of diet, nutrition and physical activity for cancer prevention. The second report also included the widely cited recommendations to eat wholegrains, vegetables, fruits and legumes, limit foods high in fat, starches and sugar and limit red and processed meat [4]. The third report released in 2018, provided updated but largely similar recommendations based on evidence from 17 cancers and data on 51 million people [5]. Evidence on diet and cancer is also synthesized by the WCRF through the Continuous Update Project that provides ongoing, comprehensive, evidence-based reports.

Despite the large global evidence base, criticism of studies on diet and cancer, particularly epidemiologic studies, persist among researchers [6] and the general public [7,8,9,10]. This reflects, in part, the divergence in results that are disappointingly common in the field [6,10]. For example, dietary fiber and carotenoids have been reported to be positively, inversely or not associated with colon cancer and lung cancer, respectively [7]. The issues and challenges that underpin the perceptions regarding evidence on diet and cancer have been reviewed in depth previously [8,9]. One potential path forward is the application of new technologies such as metabolomics, transcriptomics, proteomics, metagenomics, epigenetics and relatively more established technologies like genomics, to improve the accuracy of dietary assessment, characterize metabolic heterogeneity and study nutrients with greater specificity. For example, the lack of biomarkers for most nutrients and for dietary patterns limits the ability to observe diet–disease relationships. As well, most epidemiologic research focusses on nutrients broadly, such as selenium or vitamin D, without consideration of (or the ability to measure) specific forms that have different nutritional, metabolic and biological effects [11]. Advances in omics technology provides a new opportunity to discover biomarkers of nutrients (both generally and chemical species), food groups and dietary patterns [10] that may help to disentangle relationships. In addition, individual metabolic heterogeneity in response to diet and nutrients is well-established and reflects broad interactions with genes, proteins and metabolites. Unmeasured confounding from such interactions may contribute to inconsistent associations between dietary components and cancer. This review focusses on a multi-omics approach to the study of diet–cancer relationships as it relates to two broad areas 1) the use of omics to identify novel biomarkers of nutrients, foods or dietary patterns that may improve the precision of estimates in the study of diet and cancer and 2) the use of omics to understand how individuals respond to nutrients, foods or dietary patterns and the impact on diet–cancer relationships. We also outline challenges in diet and cancer focused multi-omics studies and future opportunities to advance the field.

Relevant original research in humans was identified with Embase and Medline/Pubmed. No limits were set on the type of literature, or dates of articles. Key search terms included systems biology, multi omics, nutrigenomics, metabolism, diet, nutrition, and cancer as well as MeSH terms: “biomarkers, cancer”, “diet, epidemiology”, “diet, genetics”, “diet, prevention and control”, in combination with any of “metabolomic”, “nutrigenomics”, “proteomics”, “proteogenomics” and “microbiome, human”. Due to the limited number of multi-omic or systems biology studies identified with this search strategy, we subsequently expanded the definition of systems biology to include studies that used an omic type in combination with a biomarker (e.g., inflammatory markers, or circulating dietary markers such as vitamins or fatty acid levels) that had relevance to cancer (e.g., the study of intermediate markers or risk factors for cancer). This review does not provide in-depth summaries of the statistical methods for integrating multi-omics datasets, or principles of omic techniques as these topics have been the subject of a number of reviews [12,13,14].

## 2. Omics Technologies

A brief introduction to the omic type(s): genome, transcriptome, metabolome, proteome and microbiome that may be part of an integrative approach to studying diet–cancer relationships is provided below.

### 2.1. Genome, Epigenome and Transcriptome 

The study of genomics, the interactions of genes with each other and with environmental exposures, has led to new fields of nutrition research; nutrigenomics and nutrigenetics. Nutrigenomics refers to the influence of nutrients on gene expression. Nutrigenetics studies the effects of gene–diet interactions on a phenotypic trait. A number of nutrient–gene interactions have been identified, some of which have implications for cancer development. For example, polymorphisms in the methylenetetrahydrofolate reductase (*MTHFR*) gene are associated with a lower risk of colorectal cancer [15,16] but only when folate, vitamins B12 and B6 intakes are adequate [17,18]. Given these findings, it is tempting to apply tailored recommendations for folate, vitamins B12 and B6 but the reality is more complex. Folate intake and folate serum levels have been associated with increased, decreased and no risk of various cancers [14] and even the same *MTHFR* polymorphism has been suggested to have divergent associations with non-Hodgkin lymphoma in different populations (positive in Caucasians and protective in Asians [19]).

DNA methylation; an epigenetic modification, involves the transfer of a methyl group to the 5’ position of cytosine. DNA methylation has been associated with a number of processes including the repression and activation of gene transcription and can be repressed via hypermethylation and hypomethylation [20]. Altered methylation patterns have been shown to promote carcinogenesis via gene transcription of oncogenes and tumor suppressor genes [21]. DNA methylation can be altered globally or at specific sites by environmental exposures, including diet [22,23,24,25]. For example, a number of studies (recently reviewed by Mahmoud and Ali [22]) have shown associations between micronutrients that can act as methyl donors or methylation co-factors [26]; choline, folate, betaine, methionine, vitamin B12 and B6, and global DNA methylation in tissues from patients with cancer.

Transcriptomics has been used to extensively profile gene transcription in cells. Nearly 800,000 gene expression datasets related to cancer are contained in the Gene Expression Omnibus database of the National Center for Biotechnology (NCBI) of the National Institutes of Health. Transcriptomics can be applied to study how diet can induce alterations in gene expression. For instance, transcriptome alterations in adipose tissue were observed following a low-calorie diet in a study of 191 obese participants. Both baseline transcriptomics and transcriptomic alterations were predictive of dietary response (i.e., weight loss and glycemic outcomes [27]) among participants. 

### 2.2. Metabolome 

Metabolomics is the measurement of small molecules in biological specimens that represent an integrated picture of upstream genetic, transcriptomic and proteomic variation. Metabolomics provides a detailed characterization of metabolic phenotypes, and corresponding metabolic derangements that may contribute to disease. The application of metabolomics has provided insight into interactions between dietary intake and metabolic pathways that may lead to cancer development. For example, epidemiologic evidence suggests diets rich in carotenoids containing fruits and vegetables are associated with a reduced risk of lung cancer [28]. However, two subsequent chemoprevention trials unexpectedly found higher lung cancer risks with beta carotene supplementation [29,30]. A metabolomics assessment of men who were randomized to beta-carotene supplements in the Alpha-Tocopherol, Beta-Carotene Cancer Prevention (ATBC) study reported 17 metabolites changed in response to beta-carotene supplementation. The findings provided the first insight into the biological effects of beta-carotene supplementation and suggested that interactions between medications and dietary supplements warrant further exploration.

### 2.3. Proteome 

The proteome encompasses all proteins expressed in a given cell, tissue, or organ. Assessments of proteomics are technically challenging and proteomic technologies are costly which is reflected in the comparatively low number of studies (with respect to other omics types) which have adopted proteomic tools in diet–cancer research.

### 2.4. Microbiome

The microbiome consists of trillions of microbes including bacteria, fungi, parasites and viruses. Genetics, diet, alcohol, medication and other environmental exposures are all potent contributors to host-microbiome interactions [31]; which include local and distal effects on immunity, inflammation, endocrine and metabolic function [32,33,34]. The microbiome has been proposed to play an etiologic role in some cancers, such as post-menopausal breast cancer via the ‘estrobolome’, a group of bacterial genes and functional pathways that metabolize estrogens and modulate estrogen homeostasis [35]. The microbiome is highly variable between individuals [36,37,38]. Recent data suggests that inter-individual variability may contribute to differences in the response to dietary interventions [39], which may have implications for disease prevention.

### 2.5. Integrative-Omics 

The integration of multiple omics technologies is increasingly recognized as a powerful tool for providing greater depth to research findings. Metagenomics can be applied to characterize and identify the microbial community, while transcriptomics, proteomics, and metabolomics can be used to understand the gene expression, protein production and metabolism of the community, respectively. The metabolic and genetic profiles of an individual are strongly linked [40] which enables the consideration of genetic factors such as polymorphisms that influence the metabolic make-up of a given person or population. The metabolome and microbiome are also complementary. Many metabolites have a short half-life and metabolomics studies may thus be enriched with the integration of metagenomics as the gut microbiota is relatively stable over time [41]. Similarly, metabolomics can help overcome some of the limitations of microbiome characterization by providing quantitative functional annotation that is not possible with 16S rRNA sequencing [42]. Using the above example of variable associations between folate intake and cancer, a consideration of the contribution of folate intake (as well as dietary intake more holistically) on gene methylation, proteomics, metabolomics and the metabolome would help researchers to fully understand the variability and complexity of these relationships.

Figure 1 depicts a simplified overview of how omics technologies could be applied to study diet–cancer relationships. 

## 3. Biomarkers, Diet and Cancer

Omics biomarkers could contribute to the improved accuracy of dietary assessments by validating consumption, identifying underreporting and assessing adherence to dietary interventions. Parallels can be drawn to studies of recovery biomarkers and biomarkers with a direct relationship between intake and tissue levels [43]. Twenty-four-hour urinary nitrogen has been used to validate estimates of protein intake [44]. Doubly labelled water (DLW) measures average energy expenditure among weight-stable individuals, and can be used as a marker of energy intake [45]. The Women’s Health Initiative used DLW to calibrate self-reported energy intake assessed by food frequency questionnaires to minimize the effects of over and underreporting. Using DLW-calibration, total energy intake was associated with increased risk of postmenopausal breast cancer; this association was not observed in the absence of calibration [46]. In addition urinary potassium and sodium have been used to calibrate self-reported intake in epidemiological studies [46,47]. However, the number of recovery biomarkers, is currently very limited and widespread application is hampered by cost (DLW) or difficulty collecting biospecimens (24-hr urine). Other biomarkers which are correlated to dietary consumption are available such as omega-3 fatty acids for seafood intake [48] and carotenoids for fruit and vegetable intake [49] but neither are directly proportional to dietary intake. 

Most of the studies that have sought to identify biomarkers of foods, nutrients and dietary patterns have focused on metabolomics [10]. Metabolomics reflect the current biological status of an individual since any alteration in metabolism/bioenergetics alters downstream biological functions including the metabolism of food in the body. A large number of associations between metabolites and diet have been identified including biomarkers of coffee consumption (e.g., diacylphosphatidylcholine C32:1 and phenylalanine), high fiber diets (e.g., 2-aminophenol sulfate and 2,6-dihydroxybenzoic acid) and Western dietary patterns (e.g., high levels of leucine, phenylalanine and short-chain acylcarnitines), as reviewed in Guasch-Ferré and Hu [10]. However, one of the limitations of a metabolomics-based approach to biomarker discovery is that metabolites can have a short half-life and may be more likely to reflect immediate term dietary intake. 

Comparably few studies have integrated other omics technologies with metabolomics to identify dietary biomarkers. Valid biomarkers of diet need to be sensitive to intake and ideally reflective of usual intake, especially for studying the role of diet in long-latency diseases like cancer. Microbiota in the gut are influenced by long-term dietary intake and shifts in the composition can also be observed with acute dietary changes [50]. Combined microbiome and metabolome approaches may therefore be an attractive approach for identifying biomarkers or providing context for identified metabolite biomarkers.

Tang and colleagues [51] applied metabolomics and metagenomics analyses to study usual dietary intake in healthy participants (Table 1). In addition to identifying associations between diet and metabolites, they conducted analyses to define inter-relationships between diet, the microbiome and metabolome and additional analyses to determine whether identified associations were mediated by microbial enterotype. Strong relationships were observed between bacterial taxon in the gut microbiome and multiple foods and nutrients from self-reported long-term diet. For example, Lachnospira was positively associated with vitamin E and Subdoligranulum and Ruminococcaceae NK4A214 were positively associated with vitamin E. Additional bacterial taxon associations were observed for alpha-carotene, beta-carotene, lutein and zeaxanthin, vegetables, vitamin B12, folate, fiber, milk, cheese, calcium, zinc, sodium, magnesium, potassium, aspartame, mannitol and trans fat consumption. Many nutrients were associated with more than one bacterial genus which may be particularly informative as multi biomarker panels can offer a better estimation of diet. In addition, associations overlapped for some nutrients (e.g., alpha and beta carotene, lutein and zeaxanthin, vegetables, vitamin E, folate, fiber, magnesium and potassium were all associated with Bacteroides), suggesting signatures were associated with dietary patterns, although dietary patterns were not directly studied for reasons that are unclear. The identification of diet–microbiome associations that reflect long term diet may be particularly relevant for studying the downstream effects of diet on long latency chronic diseases such as cancer. Further analysis revealed microbiome mediated effects on metabolic pathways particularly bile acid, amino acid and xenobiotic signaling, which is interesting given the importance of bile acids in the etiology of some cancers [52,53].

While the above study by Tang and colleagues [51], was modest in scale (N = 75 with plasma and stool samples), it is one of the largest studies to study relationships between diet, the microbiome and metabolome. This underscores the need for further studies, particularly the replication and validation of biomarkers in larger populations. In 2018, the National Institutes of Health convened a 2-day workshop on the use of multi-omics for the discovery of nutritional biomarkers [54]. The recognition of the dearth of multi-omic studies and opportunities for novel nutritional biomarker discovery by the major research funding body in the United States will hopefully spur progress in the field. The resulting publication from the workshop also eloquently highlights the many challenges for diet biomarker discovery including the need for more data on dietary patterns, the food metabolome, the establishment of dynamic ranges of biomarkers, and the need for standardized collection, analysis and reporting of biospecimens and biomarker data to facilitate study replication [54].

## 4. Response to Diet

Relationships between diet and cancer may be seen with greater clarity if variation within a population can be identified. Stratification of populations with different risks of cancer or characteristics such as polymorphisms that affect the metabolism of dietary components may help to contextualize results as well as to focus studies on select populations. Moreover, biomarkers of diet such as metabolites may represent the bioavailable dose and could provide insight into variable responses in the absence of measures such as polymorphisms. Variation in the response to diet may also arise artificially due to a failure to distinguish between forms of micronutrients that have different biological properties (e.g., selenomethionine, selenocysteine; two forms of selenium) [10]. The ‘true’ response to diet may be observable if omics technologies are applied to enhance the precision with which dietary components are assessed.

Omics approaches may be particularly valuable for risk-relationships where there is uncertainty i.e., evidence for the association between dietary patterns or components and cancer is ‘limited’ such as foods containing vitamin D and vitamin D supplements [59]. Both Kazemian [56] and Lowe [57] studied whether variation in vitamin D was related to vitamin D receptor (VDR) polymorphisms. Kazemian et al. reported changes in intermediate markers of cancer (i.e., inflammatory factors) pre-post vitamin D supplementation in breast cancer survivors differed by genotype. Lowe et al. measured a biomarker of vitamin D status (serum 25(OH)D) and polymorphisms in vitamin D receptors (VDR) among breast cancer cases and controls. They reported increased odds of breast cancer in women with a polymorphism in VDR and an additive effect of low 25(OH)D on odds of breast cancer. The response to dietary intake was also studied by Citronberg et al. [58], albeit with an aim that differed markedly from Kazemian and Lowe. In this study the association between dietary intake, inflammatory markers (lipolysaccarides (LPS) and C-reactive protein) and the gut microbiome were studied to gain insight into mechanisms through which the microbiome may contribute to obesity and chronic diseases like cancer. They found that saturated fat intake was positively associated with LPS and additionally reported that bacteria phyla and the abundance of bacteria (e.g., actinobacteria) differed by LPS concentration. Although the study was cross-sectional, the findings provide evidence of dietary shifts in the gut microbiome that may be relevant for cancer development.

A natural extension of characterizing the individual response to diet is personalized nutrition or precision nutrition that aims to develop individually tailored nutrition intervention and/or advice for an individual which is known to be beneficial [60]. Although promising, much more work is needed, especially when health outcomes like cancer are considered; thus we have not included a discussion of this topic. However, interested readers are directed to the study by Zeevi et al. [39] described in Table 1 and below, which included a personalized nutritional intervention among the other aims which are discussed in this review. We further suggest that the incorporation of omics approaches will be necessary to achieve the lofty goals of precision nutrition.

Zeevi et al. [39] conducted a study characterizing the glycemic response to meals in a cohort of 800 in addition to gut microbiota (and other parameters). Although the study was framed towards prediabetes and type 2 diabetes, it is noteworthy due to the novel study design and potential relevance of blood glucose to cancer [61] and as such we have included it here. They reported high variability in postprandial glucose response (PPGR) between participants after consumption of the same standardized meal as well as after similar self-selected meals were consumed. The variability in response could be attributed in part to differences in the microbiome, and in particular to Proteobacteria and Enterobacteriaceae which are associated with glycemic control. They also successfully altered the microbiota composition and attenuated PPGR with a dietary intervention, although whether the alteration was lasting or translates into the prevention of chronic disease was not possible to ascertain. 

## 5. Challenges and Future Opportunities

Multi-omics approaches hold considerable promise for advancing nutrition research. However, study design considerations and challenges need to be addressed in order to improve studies of diet–cancer relationships. Considerations should include choice of tools/technology for both self-reported diet and omics measurement, population selection, and approaches for integrating omics data. Choosing the most appropriate self-reported dietary assessment tool(s) for a given study design and study aim(s), as well as using appropriate statistical methods for the analysis of self-reported dietary data, is an important consideration to minimize the error of estimates [46,62,63,64]. This is applicable to all diet-based studies and particularly relevant for biomarker discovery which relies on a reasonably accurate assessment from self-reported diet. To date, many studies have focused on single nutrients or foods versus dietary patterns which is a disconnect with the reality that diet as a whole is more than the sum of its parts [65]. With respect to omics approaches, choice of the source(s) of biological samples, sample preparation methods, analytic platform (e.g., nuclear magnetic resonance or liquid chromatography-mass spectroscopy for metabolomics), impose various strengths and limitations on the measurement and interpretation of data as summarized previously [66]. The omics approach also needs to be matched to the study aim. For example, usual dietary intake may be less amenable to metabolomics and proteomics approaches as metabolites and proteins which tend to have a short half-life [67]. Conversely, the gut microbiome in adults is relatively stable over time [41,68] and integration with other omics methods may be more appropriate for studying usual intake.

Population selection is another important aspect. The identification of subgroups that vary in their response to diet requires large, diverse populations in different regions of the world, particularly in the case of rare polymorphisms. Large samples with a reasonable range and variation in dietary intake are also needed to identify robust biomarkers and importantly, validate measurements, as currently few biomarkers of diet have been validated [54]. Larger feeding studies that include a variety of foods and dietary patterns are necessary to spur the identification of biomarker candidates. The field would also greatly benefit from consortium-based approaches that combine prospective cohort studies and foster collaborative approaches to study common aims. The Consortium of Metabolomics Studies (COMETS) [69] brings together 47 prospective cohort studies with blood metabolomics, many of which have additional omics data and dietary data which could be leveraged. The National Cancer Institute (NCI) Cohort Consortium [70] was founded to undertake large-scale pooling research. It consists of 58 epidemiologic cohorts from 20 countries, 9 million participants, of whom 2 million have biospecimens. Although to our knowledge, there are no consortium projects with aims that match the topics discussed in this review.

The integration of different omics data types is an ongoing challenge [66]. Temporality, that is assumptions about the direction of temporal effects, need to be made, with the exception of genetics where it is reasonable to expect genetic variation precedes most other omic measurements. A multitude of statistical approaches for integration of multi-omics and data reduction techniques for hundreds to thousands of metabolites, proteins, and bacteria have been utilized, with no clear best approach(es). For example, pathway mapping provides insight into the biological relevance of findings [71,72] but the statistical power of integrative analyses may not be fully realized [73]. Identifying the best approaches would be a complex endeavor but would nonetheless help facilitate progress in the field and comparability across studies. Otherwise, the potential for new insights from integrative approaches will be harder to realize.

Of note, even with advancements in omics and diet, there remains the need to include traditional self-reported methods of dietary intake. Studies are likely to uncover concentration biomarkers; indirect measures of intake that reflect the concentration of a chemical or compound that is subject to individual variation in metabolism rather than true intake which is the aim of self-reported dietary assessment [43]. Biomarkers do not provide contextual factors related to dietary consumption that is captured in self-reporting, they tend to be expensive, and as of yet have not been shown to be sensitive or specific to intake [54]. Combined approaches to dietary assessment which use food frequency questionnaires (FFQs), together with 24-hr recall or food diaries can reduce the error in estimation [63], however the burden on participants is greater and response rates decline over subsequent collection time points [64]. This burden is potentially problematic when coupled with the need for biospecimen collection and the cost of dietary assessment and omics in large epidemiologic studies. Indeed, a prior review of dietary tools found that tools with low cost and participant burden are often used at the expense of data quality [74]. Strategies to identify and promote the application of best practices in dietary assessment and omics with an emphasis on dietary patterns versus single foods/nutrients, are needed and would benefit from collaborative approaches. This is a critical part of advancing the field of diet and cancer, as clarity cannot be expected from omics technologies even among large populations in the absence of accurately measured diet.

## 6. Conclusions

This review revealed the scarcity of multi-omic diet–cancer studies in humans. Indeed, the limited evidence base resulted in broadening the scope of studies discussed to those that included at least a biomarker in combination with an omic technology, and those that had relevance to cancer even if cancer was not part of the central study design. Despite this, the number of studies captured was limited and it was not possible to identify foods or nutrients that may be particularly amenable to multi-omics approaches to clarify diet–cancer associations. This reflects the relatively recent advent of high throughput technology, evolving tools (statistical methods, software) for the integration of complex genomics, metagenomics, proteomics and metabolomics data, and the cost of omics technologies that limit multi-omics measurement and implementation in larger populations, among other challenges. However, this emerging field of research, holds great promise for advancing the understanding of diet and cancer by providing a novel insight into how variation in the response to diet can impact cancer-related pathways and improve the assessment of dietary exposures. Accomplishing these goals will provide a clearer picture of the relationship between diet and cancer but collaborative approaches that address challenges and refine approaches are needed for any advancements to be realized.

## Figures and Tables

**Figure 1 metabolites-10-00123-f001:**
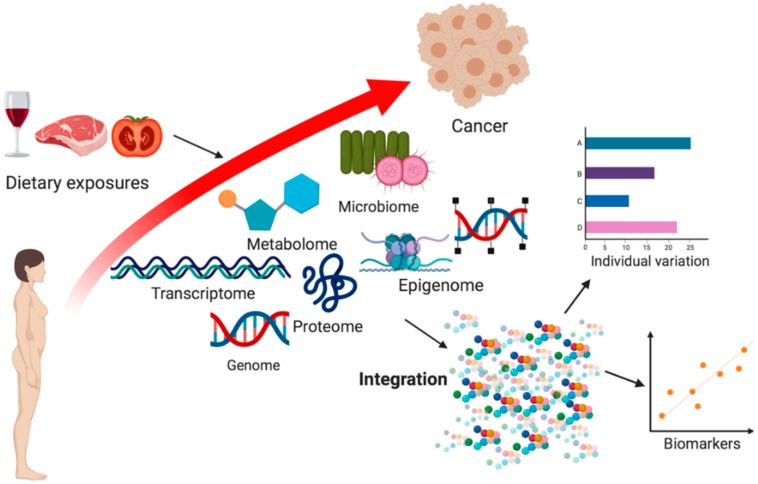
Systems biology approach to studying the biological processes linking diet and cancer. Through integration of omics approaches spanning the genome, epigenome, transcriptome, metabolome, proteome and microbiome, biomarkers of dietary exposures and the variation in the biological response to diet may be identified. Created with Biorender.com.

**Table 1 metabolites-10-00123-t001:** Key studies using multi-omics or omics in combination with a biomarker to study diet–cancer relationships.

Author	Population	Aim	Omics or Biomarkers Assessed	Dietary Measures	Key Findings
Zeevi et al. [39]	800 healthy subjects aged 18–70	To measure individualized post prandial glucose response, variability in response and factors related to variability	Microbiome in stool (16S rRNA), blood glucose	Food frequency questionnaires at baseline, food logs, and standardized meals provided to subjects	High interpersonal variability in post prandial glucose response to the same meal (self-reported and standardized). Variability was associated with microbiome taxa and phylum. Microbiome factors were used (along with other factors) to predict post prandial glucose response
Tang et al. [51]	136 healthy subjects	To determine associations between microbiome composition and habitual diet	Microbiome (16S rRNA) in stool and saliva, metabolomics (plasma and stool) on a subset (N = 75)	3-day food records and food frequency questionnaire (NCI’s DHQ I) [55]	On a global level, long-term diet was associated with the gut microbiome while short-term diet was associated with the gut and plasma metabolome. 61 dietary nutrients, predominately plant and dairy derived were associated with at least 1 bacterial genus. Metabolic flux through plant-derived nutrients and metals were susceptible to interactions between diet and microbiome composition
Kazemian et al. [56]	Single-arm non-randomized pre-and post-trial in 176 breast cancer survivors who received vitamin D supplementation for 12 weeks	To study if polymorphisms in vitamin D receptor (VDR) are associated with change in biomarkers known to relate to breast cancer risk and survival	VDR polymorphisms, Biomarkers: E-cadherin, MMP9, interferon β, s-ICAM-1, VCAM-1, TNFα, IL6, PAI-1, hs-CRP	4000 IUD vitamin D3 supplement, 3 X 24-hr food records	Variation in the response to vitamin D supplementation was observed. Changes in cancer biomarkers pre and post vitamin D supplementation differed by genotype and haplotype, e.g., women with AA and GA genotypes of cdx2 had greater increase in MMP9 levels. Genotype differences were also observed for TNFα, suggesting potential relevance for cancer risk and survival.
Lowe et al. [57]	Breast cancer patients (n = 179) and control women (n = 179) in the United Kingdom	To determine whether low 25(OH)D levels in combination with VDR polymorphisms are associated with breast cancer risk	VDR polymorphisms, Biomarker: plasma 25(OH)D levels measured by ELISA	None in addition to plasma 25(OH)D	25(OH)D levels were lower in breast cancer patients. Increased odds of breast cancer among people with the BsmI polymorphism in the VDR. People with both both low 25(OH)D and BsmI polymorphism had the greatest risk of breast cancer while those with either low 25(OH)D or BsmI had intermediate risk.
Citronberg et al. [58]	110 premenopausal women in the United States	To dermine associations between gut microbial communities, inflammation and dietary intake	Microbiome in stool (16S rNA), Biomarkers: plasma LPS-binding protein and CRP	3-day food records	Dietary fat intake, particularly saturated fat intake and CRP were positively associated with LBP. The abundance of actinobacteria and lipopolysaccharide synthesis differed by LPS tertile.

Abbreviations: hs-CRP; human high sensitivity C-reactive protein; ICAM; intercellular adhesion molecule-1, IFN; interferon, IL; interleukin, LBP; lipopolysaccharides binding protein; LPS; lipopolysaccharides, MMP9; matrix metallopeptidase, PAI; plasminogen activator inhibitor; TNF; tumor necrosis factor, VDR; vitamin D receptor.

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
