# Peer review of "An Integrative Approach to Assessing Diet–Cancer Relationships"

_metabolites, 2020, doi:10.3390/metabo10040123_

Round 1

Reviewer 1 Report

A very interesting review about an integrative approach to assessing diet-cancer relationships.  

I have some suggestions : 

Line 51 : which types of cancer are, at present, more interested by multi-omic studies (to investigate the relationship between diet and desease) ? . Please , provide more details.

Line 187 : which foods are, at present, more involved in the relationship between cancer and diet and , in particular , in multi-omic studies ?. Please, provide more details.

Author Response

Line 51 : which types of cancer are, at present, more interested by multi-omic studies (to investigate the relationship between diet and disease)? Please, provide more details.

-Multi-omics studies can provide insight into the relationship between diet and all types of cancer, although they may be particularly useful for clarifying risk relationships between dietary components/patterns and cancer that have been identified as needing more evidence to confirm and/or grade the strength of the recommendation. We have added the following to the text: “Omics approaches may be particularly valuable for risk-relationships where there is uncertainty i.e. evidence for the association between dietary patterns or components and cancers is ‘limited’…”

Line 187 : which foods are, at present, more involved in the relationship between cancer and diet and, in particular  in multi-omic studies? Please, provide more details.

-We are not sure that we fully understand the reviewer’s question. The study of diet and cancer is a huge body of evidence that encompasses many single nutrients, food groups and dietary patterns. The strength of the evidence linking diet and cancer varies and has been summarized in the comprehensive reports from the World Cancer Research Fund International (which is referenced in our study). With respect to multi-omic studies, there are too few to make any generalizations about which foods or other aspects of diet are more (or less) strongly related to cancer.

Reviewer 2 Report

Significant Contribution to the Field?

This is a review that attempts to sort out the potential of multi-omics approaches to ferret out the links (or not) of diet to cancers.  This review appears to have a decent balance of topics and brings the microbiome aspects to the front, indeed that appears to be one of the goals.

However, a major dietary category with numerous high-profile clinical trials, namely essential elemental micronutrients such as Se are not even mentioned, despite a Se-specific reference being cited (ref 6) – this citation is an editorial, not a peer-reviewed paper, about the EPIC report – but the point is that the author fails to discuss both the cited subject (Se) and some very major lessons pointed out by the cited editorial.  The cited editorial is of substantial significance in that it points out a fundamental flaw in clinical trials of dietary Se:  the chemical forms are usually not known, or that a questionable chemical form (this editorial does not specifically point this out) was used in the giant SELECT trial due to a miscarriage of due diligence.  Chemically and biochemically, discussion of dietary “selenium” (the element) and cancer is akin to relating dietary “carbon” (the element) and cancer – it is a complete waste of time.  The author fails to even mention such extremely significant failings about the majority of prior studies in micronutrients [lack of] chemistry & biochemistry, metabolites, and cancer.

Related to this topic, the author has limited part of the review to “metabolomics” which is correctly stated (line 84) as the “…is the measurement of small molecules in biological specimens…” but fail to delineate that metabolomics (the analytical activity) is NOT a measure of metabolism (e.g. metabolic pathways or biochemistry); unfortunately the remainder of that paragraph and throughout the article, implicitly (optimistically?) equate the two.

Continuing on using the examples above, the ref 6 is an editorial about a paper (EPIC study) in the same issue that sloppily looked at total Se element, but to the paper’s credit also segregated out Se incorporation into proteins (selenoprotein P, SePP).  This is another huge missed opportunity for this review:  macromolecules are also “metabolites” and conceptual barriers artificially imposed by technological barriers such as measuring “small molecules” (line 84), are a huge disconnect that sabotages useful discussion of diet and cancer.  Selenium is a long-standing example (out of many) of intellectual shortcomings of such studies, but there is no hint of recognition of this huge problem which was described right under the nose (it was cited) of the author.

On the other hand, the discussion of “biomarkers” appears to be well-reined in, generally avoiding over-interpretation regarding the biology.  A bit of a shortcoming is that “biomarkers” of diet & cancer is such a super-giant field, that it is difficult to do justice covering it in a single review; the author does an adequate job (feels like a retreat…..) by calling for coordination such as identifying “best practices” and the like (section on Challenges).  There is bit of disturbing faith implicit in that section, which is that sufficiently large studies (good stats) will prevail, but this may represent another missed opportunity of the review:  it should be considered that if mis-concepts and intellectual failings direct the analytical technology and study design, ever larger studies measuring the wrong things will NOT prevail.  And they have not.

Organized and Comprehensively Described?

It is adequate, but “comprehensive” is not possible in such a short review, so that is an unfair metric.  No doubt many readers will be disappointed in the review which did not, and cannot, cite thousands of papers.

Scientifically Sound and Adequate References?

It is reasonably sound, but there are missed opportunities as outlined above.  As for “adequate” references, again it is not possible to cite even a small fraction of the body of work in some areas such a biomarkers, so this reviewer is compelled to support the author’s practice of “cherry-picking” citations (which in some other contexts should be stricken).

English correct and readable?

This is fine.

Author Response

This is a review that attempts to sort out the potential of multi-omics approaches to ferret out the links (or not) of diet to cancers.  This review appears to have a decent balance of topics and brings the microbiome aspects to the front, indeed that appears to be one of the goals.

However, a major dietary category with numerous high-profile clinical trials, namely essential elemental micronutrients such as Se are not even mentioned, despite a Se-specific reference being cited (ref 6) – this citation is an editorial, not a peer-reviewed paper, about the EPIC report – but the point is that the author fails to discuss both the cited subject (Se) and some very major lessons pointed out by the cited editorial.  The cited editorial is of substantial significance in that it points out a fundamental flaw in clinical trials of dietary Se:  the chemical forms are usually not known, or that a questionable chemical form (this editorial does not specifically point this out) was used in the giant SELECT trial due to a miscarriage of due diligence.  Chemically and biochemically, discussion of dietary “selenium” (the element) and cancer is akin to relating dietary “carbon” (the element) and cancer – it is a complete waste of time.  The author fails to even mention such extremely significant failings about the majority of prior studies in micronutrients [lack of] chemistry & biochemistry, metabolites, and cancer.

-Thank you for raising this. There are many limitations of trials such as SELECT and ATBC that are worthy of and have been subject to more in depth discussions. The limitations of such clinical trials and studies similar to the EPIC one mentioned below are beyond the scope of this paper. Frankly, the chemical form used in SELECT is just one of the issues in that trial, and to adequately discuss this and failings of prior studies would distract from the main purpose of the paper, and likely be better suited to a commentary versus review article. However, I agree that the widely used approach of lumping different isoforms of nutrients into one category is problematic, and one that could be potentially refined/resolved through omics approaches. The introduction has been revised starting at line 39 to reflect your comments. As well as the first paragraph in ‘Response to diet’

“One potential path forward is the application of new technologies such as metabolomics, transcriptomics, proteomics, metagenomics, epigenetics and relatively more established technologies like genomics, to improve the accuracy of dietary assessment, characterize metabolic heterogeneity and study nutrients with greater specificity. For example, the lack of biomarkers for most nutrients and for dietary patterns limits the ability to observe diet-disease relationships. As well, most epidemiologic research focusses on nutrients broadly, such as selenium or vitamin D, without consideration of (or the ability to assess) specific forms that have different nutritional, metabolic and biological effects (11). Advances in omics technology provides a new opportunity to discover biomarkers of nutrients (both generally and chemical species), food groups and dietary patterns (10)that may help to disentangle relationships.”

“Variation in the response to diet may also arise artificially due to failure to distinguish between forms of micronutrients that have different biological properties (e.g. selenomethionine, selenocysteine; two forms of selenium) (10). The ‘true’ response to diet may be observable if omics technologies are applied to enhance the precision with which dietary components are assessed.“

Related to this topic, the author has limited part of the review to “metabolomics” which is correctly stated (line 84) as the “…is the measurement of small molecules in biological specimens…” but fail to delineate that metabolomics (the analytical activity) is NOT a measure of metabolism (e.g. metabolic pathways or biochemistry); unfortunately the remainder of that paragraph and throughout the article, implicitly (optimistically?) equate the two.

-It was not the intention to conflate the two. I have clarified this in the description of metabolomics “Metabolomics provides detailed characterization of metabolic phenotypes, and corresponding metabolic derangements that may contribute to disease.” I have also edited the text throughout the manuscript to ensure we accurately reflect what is captured by metabolomics.

Continuing on using the examples above, the ref 6 is an editorial about a paper (EPIC study) in the same issue that sloppily looked at total Se element, but to the paper’s credit also segregated out Se incorporation into proteins (selenoprotein P, SePP).  This is another huge missed opportunity for this review:  macromolecules are also “metabolites” and conceptual barriers artificially imposed by technological barriers such as measuring “small molecules” (line 84), are a huge disconnect that sabotages useful discussion of diet and cancer.  Selenium is a long-standing example (out of many) of intellectual shortcomings of such studies, but there is no hint of recognition of this huge problem which was described right under the nose (it was cited) of the author.

-Macromolecules were not considered in this review since they are not widely recognized as metabolites, a term usually restricted to small molecules. I appreciate the reviewer’s concern that this is an artificially imposed barrier, however, I believe that the consistent use of the term metabolites to reflect analytes from metabolomics studies across papers is needed to minimize confusion. The aim of this project was to emphasize the need to move towards an integrated systems approach which considers the many hundred to thousand markers measured by omics technologies. While it is clear that consideration and measurement of macromolecules is a critical area in diet-cancer (and more broadly diet-disease) relationships, we are unclear how macromolecules are integrated into systems biology approaches, and the flow of information from an environmental exposure through to phenotype. I have however, revised the manuscript as outlined above to recognize the importance of macromolecules.  

On the other hand, the discussion of “biomarkers” appears to be well-reined in, generally avoiding over-interpretation regarding the biology.  A bit of a shortcoming is that “biomarkers” of diet & cancer is such a super-giant field, that it is difficult to do justice covering it in a single review; the author does an adequate job (feels like a retreat…..) by calling for coordination such as identifying “best practices” and the like (section on Challenges).  There is bit of disturbing faith implicit in that section, which is that sufficiently large studies (good stats) will prevail, but this may represent another missed opportunity of the review:  it should be considered that if mis-concepts and intellectual failings direct the analytical technology and study design, ever larger studies measuring the wrong things will NOT prevail.  And they have not.

-I agree with the reviewer. I have refined our wording in the ‘considerations’ section as well as the conclusion section to reflect that without strategies to address the challenges, the field will not move forward. For example “Accomplishing these goals will provide a clearer picture of the relationship between diet and cancer but collaborative approaches that address challenges and refine approaches are needed for any advancements to be realized.” And “This is a critical part of advancing the field of diet and cancer, as clarity cannot be expected from omics technologies even among large populations in the absence of accurately measured diet.“

Organized and Comprehensively Described?

It is adequate, but “comprehensive” is not possible in such a short review, so that is an unfair metric.  No doubt many readers will be disappointed in the review which did not, and cannot, cite thousands of papers.

-I have now addressed this in the paper. The most comprehensive source related to diet and cancer (although not specific to omics) remains the WCRFI report as they have dedicated resources to continually update the evidence base. The goal was to focus specifically on studies which have used more than one omics approach to answer questions related to diet and cancer and not to provide a comprehensive review of single omics studies of diet and cancer or diet and cancer more generally since as the reviewer points out, this is not possible in a single review paper.

Scientifically Sound and Adequate References?

It is reasonably sound, but there are missed opportunities as outlined above.  As for “adequate” references, again it is not possible to cite even a small fraction of the body of work in some areas such a biomarkers, so this reviewer is compelled to support the author’s practice of “cherry-picking” citations (which in some other contexts should be stricken).

-I have addressed the opportunities highlighted by the reviewer as outlined above

Reviewer 3 Report

In this manuscript, the author reviews system biology approach to uncover links between diet and cancers. The review is written in a clear fashion; however, it lacks direction and completeness. Here are a few suggestions to makes the review more valuable for the readership of the journal.

- “To this end, advances in omics technology is recognized as a new opportunity to discover biomarkers.”  Can the author give example, if any, of omics biomarkers representative of dietary patterns?

- Detail how metabolomics could overcome reporting issue and individual variability in response to diet/nutrients.

- What does “microbiomics” refers to? Did the author mean metagenomics? If so, metagenomics should be used throughout.

- The following statement needs clarification. What were the initial search terms, what could explain the lack of studies found?

 “Due to the limited number of multi-omic or systems biology studies in our area of focus, we 
 expanded the definition of systems biology to include studies that used an omic approach in combination with a biomarker (e.g. inflammatory markers, or circulating dietary 
markers such as vitamin or fatty acid levels) that had relevance to cancer (e.g. study of intermediate 
markers or risk factors for cancer). 

-The “Omics technologies” section lacks structure and is incomplete. It would be easier to read if the author would describe one technology at the time and show the example of how it was uses in the diet-cancer context. Moreover, microbiota could be added to Figure 1.

-What about epigenomics? There are many studies putting in DNA methylation, diet and cancer in relation (for example: PMC6471069/)

-Most of the studies presented by the author focus on single micronutrient and not dietary patterns. This could be further discussed.

- Kazemian et al. and Lowe et al. do not seem to have assessed dietary habits or concluded anything from diet other than vitamin D intake. How do these studies qualify as omics to uncover diet-cancer relationship?

-It would be interesting if the author made suggestions/comments on how to design multi-omics experiments to better investigate diet effect in cancer.

Author Response

“To this end, advances in omics technology is recognized as a new opportunity to discover biomarkers.”  Can the author give example, if any, of omics biomarkers representative of dietary patterns?

We have revised this sentence to “To this end, advances in omics technology are recognized as a new opportunity to discover biomarkers of single nutrients, food groups and dietary patterns.” In the section ‘Biomarkers, diet and cancer’, we provide examples of metabolomic biomarkers as suggested. “Indeed, a large number of potential biomarkers have been identified including biomarkers of coffee consumption (e.g. diacylphosphatidylcholine C32:1 and phenylalanine), a high-fiber diet (e.g. 2-aminophenol sulfate and 2,6-dihydroxybenzoic acid) and a Western dietary pattern (e.g. high levels of leucine, phenylalanine and short-chain acylcarnitines), as reviewed in Guasch-Ferré and Hu.” Please also see our comment below regarding dietary patterns.

Detail how metabolomics could overcome reporting issue and individual variability in response to diet/nutrients.

-We have clarified how omics could overcome each of these issues “Omics biomarkers could contribute to improved accuracy of dietary assessments by validating consumption, identify underreporting and assessing adherence to dietary interventions. Parallels can be drawn from studies of recovery biomarkers, biomarkers with a direct relationship between intake and tissue levels (42). 24-hr urinary nitrogen has been used to validate estimates of protein intake (49)…”. And “Relationships between diet and cancer may been seen with greater clarity if variation within a population can be identified. Stratification of populations with different risk of cancer or characteristics such as polymorphisms that affect the metabolism of dietary components may help to contexualize results as well as focus studies to select populations. Whatsmore, biomarkers of diet such as metabolites may represent the bioavailable dose and could provide insight into variable response in the absence of measures such as polymorphisms.“  

What does “microbiomics” refers to? Did the author mean metagenomics? If so, metagenomics should be used throughout.

-We have replaced the term throughout the manuscript.

The following statement needs clarification. What were the initial search terms, what could explain the lack of studies found?

“Due to the limited number of multi-omic or systems biology studies in our area of focus, we 
expanded the definition of systems biology to include studies that used an omics approach in combination with a biomarker (e.g. inflammatory markers, or circulating dietary markers such as vitamins or fatty acid levels) that had relevance to cancer (e.g. study of intermediate markers or risk factors for cancer). 

-We have now included details on our search terms. “Relevant original research in humans was identified with Embase and Medline/Pubmed. No limits were set on the type of literature, or dates of articles. Key search terms included systems biology, multi omics, nutrigenomics, metabolism, diet, nutrition, and cancer as well as MeSH terms: “biomarkers, cancer”, “diet, epidemiology”, “diet, genetics”, “diet, prevention and control”, in combination with any of “metabolomic”, “nutrigenomics”, “proteomics”, “proteogenomics” and “microbiome, human”.

As we mention in the discussion (Lines 567-570), we believe the limited number of studies “reflects the relatively recent advent of high throughput technology, evolving tools (statistical methods, software) for integration of complex genomics, metagenomics, proteomics and metabolomics data, and cost of omics technologies that limit measurement of multi-omics and implementation in larger populations, among other challenges.” We expect the number of studies that would be captured in the coming years will increase exponentially

The “Omics technologies” section lacks structure and is incomplete. It would be easier to read if the author would describe one technology at the time and show the example of how it was uses in the diet-cancer context. Moreover, microbiota could be added to Figure 1.

-Please see our response to the comment below. We have added the microbiome to Figure 1 as suggested.

What about epigenomics? There are many studies putting in DNA methylation, diet and cancer in relation (for example: PMC6471069)

-The reviewers comment is well-taken. The ‘omics technologies’ section was meant to provide a high level overview of singular omics approaches as an introduction to the deeper discussion of integrative omics for diet and cancer. However as reflected in this comment and the one above, this approach inadvertently resulted in a truncated presentation. We have re-structured this section as suggested and have added information about technologies that were previously lacking: epigenomics, transcriptomics and the microbiome.

Most of the studies presented by the author focus on single micronutrient and not dietary patterns. This could be further discussed.

-We agree that this warrants further discussion and have included mention of this point throughout the paper e.g in the discussion of the study by Tang et al. “In addition, associations overlapped for some nutrients (e.g. alpha and beta carotene, lutein and zeaxanthin, vegetables, vitamin E, folate, fiber, magnesium and potassium were all associated with Bacteroides), suggesting signatures are associated with dietary patterns, although dietary patterns were not directly studied for reasons that are unclear.” as well “To date, many studies have focused on single nutrients or foods versus dietary patterns which is a disconnect with the reality that diet as a whole is more than the sum of its parts (63).” And “Strategies to identify and promote the application of best practices in dietary assessment and omics with an emphasis on dietary patterns, are needed…”

We have also taken care to mention data on dietary patterns where it exists e.g. the paragraph on biomarkers of dietary patterns now includes an example of metabolites associated with Western dietary patterns.

Kazemian et al. and Lowe et al. do not seem to have assessed dietary habits or concluded anything from diet other than vitamin D intake. How do these studies qualify as omics to uncover diet-cancer relationship?

-We have clarified that ‘diet’ was inclusive of studies of single nutrients, food groups and dietary patterns. Although there is an emphasis in nutrition to shift from a reductionist approach to consideration of the diet holistically, the reductionist view of foods is still very important. For example, it is important to understand how nutrients such as vitamin D act in humans with respect to metabolism, on genes, hormones etc. Vitamin D in particular may be important to study in a reductionist framework as it has been studied extensively with respect to cancer risk with no clear consensus on risk-relationships. 

It would be interesting if the author made suggestions/comments on how to design multi-omics experiments to better investigate diet effect in cancer.

-Thank you for the suggestion. We agree that this is an important issue to consider. We had discussed aspects of study design in the ‘challenges and opportunities’ section e.g. choice of biological samples, sample preparation method, analytic platform etc. However, the reviewer’s point is well taken that it perhaps was not obvious these were recommendations for study design. As a result we have revised this section considerably.

Round 2

Reviewer 2 Report

Author's reply and edits are satisfactory.  This reviewer still feels there are some missed opportunities to correct misconceptions, but those are probably not worth holding up the manuscript for.

Author Response

We thank the reviewer for their comments and guidance with the prior review comments which have made this a stronger manuscript. 

Reviewer 3 Report

We still don't see the link between diet-cancer interactions in Figure 1. This figure is do general and not adapted to the topic of the review.

Author Response

We still don't see the link between diet-cancer interactions in Figure 1. This figure is do general and not adapted to the topic of the review.

- Figure 1 was previously revised to incorporate the microbiome as suggested by the Reviewer who had not asked for the figure to be specific to diet-cancer. The intent was to show readers how multi-omics approaches can inform each other as part of our general introduction to each of the technology. However, the point is well taken that it should be specific to diet and cancer and a new figure has been made.